# Genetic Basis of Gap Formation Between Migrating *Helicobacter pylori* Colonies in Soft Agar Assays

**DOI:** 10.3390/microorganisms13051087

**Published:** 2025-05-07

**Authors:** Yasmine Elshenawi, Skander Hathroubi, Shuai Hu, Xiaolin Liu, Karen M. Ottemann

**Affiliations:** Department of Microbiology and Environmental Toxicology, University of California, Santa Cruz, CA 95064, USA; yelshena@ucsc.edu (Y.E.); shuai_hu@dfci.harvard.edu (S.H.); xliu241@ucsc.edu (X.L.)

**Keywords:** *Helicobacter pylori*, soft agar, chemotaxis, motility, clonal competition, quorum sensing, outer membrane proteins, hypothetical proteins

## Abstract

*Helicobacter pylori* is a motile bacterial pathogen that causes severe gastric diseases. *H. pylori* motility and chemotaxis are key colonization factors. Motility and chemotaxis are studied in many microbes, including *H. pylori*, using soft agar assays. In these assays, bacteria are inoculated into low-percentage agar and expand in a motility- and chemotaxis-dependent manner. *H. pylori* similarly expands in soft agar, but, if a plate was inoculated at multiple points, the expanded *H. pylori* colonies did not merge and left gaps. The basis of these gaps was unknown. We report here that gap formation was not affected by media components such as nutrient and agar concentrations, nor did it require chemotaxis, but it did rely on quorum sensing. To broaden our understanding of this *H. pylori* property, an *H. pylori* Tn7 transposon library was screened for mutants that lost gap formation. Fourteen mutants were identified, with transposon sites mapped to genes encoding outer membrane proteins, cysteine-rich proteins, phosphatidyl glycerophosphate synthase, an endorestriction nuclease, and several hypothetical proteins. Our results suggest that *H. pylori* may use specific proteins to avoid contact with other *H. pylori*, a behavior that may relate to previous observations that different *H. pylori* strains do not mix populations in stomach glands.

## 1. Introduction

Bacterial motility is a widespread trait found in over 50% of bacterial species and has been shown to confer fitness advantages in diverse settings [1]. Bacterial motility is often studied using bacterial media with a low percentage of agar, so-called soft or semi-solid agar plates [2,3]. After inoculation, bacteria migrate outward from the initial point, creating an expanded colony whose formation depends on growth, motility abilities, and chemotaxis responses [2,4,5]. Soft agar migration rates are influenced by both bacterial properties, such as chemotaxis, motility, and growth, as well as media physical properties, including agar concentration [6] and nutrient availability [4,5].

In soft agar motility assays, bacteria migrate outwards from the point of inoculation, resulting in colony expansion. Often these assays are performed with several inoculations per plate, and, in this case, the expanding colonies can be in close proximity to one another. When this happens, bacterial species show varying responses. In some cases, colonies merge or cross over each other, while others produce gaps with low bacterial numbers [5,7]. Three prominent theories have been proposed to explain soft agar bacterial colony gaps: (1) gaps form because the migrating bacteria deplete nutrients in front of the colony, resulting in a poor ability to grow in the region between the colonies [5]; (2) gaps form because the migrating colonies induce mechanical stress in the agar that in turn creates physical barriers, e.g., thickened agar, that cannot be crossed by a competing colony [6]; and (3) gaps form because the colonies produce inhibitory molecules that arrest the growth of competing colonies [7]. Our initial studies that frame this work showed that *Helicobacter pylori* colonies form visible gaps in motility assays, so this work focusses on them.

In some bacterial species, soft agar colony gaps are formed only between non-clonal strains. *Proteus mirabilus,* for example, forms gaps only between non-clonal strains [8]. This response has been attributed to a genetic determinant mechanism to distinguish and recognize non-kin cells [9]. Clonal discrimination behaviors have been identified in various growth conditions including and outside of soft agar for *Bacillus subtilis* [5,7], *Pseudoalteromonas haloplanktis* [5], *Salmonella* Typhimurium [5], *Escherichia coli* [5,7], *Paenibacillus dendritiofrmis* [7], *Pseudomonas putida* [6], *Vibrio cholerae* [10] and *H. pylori* [11]. Altogether, it is clear that numerous reasons can underly gap formation in soft agar.

*H. pylori* is a motile microbial pathogen that is frequently studied in soft agar motility assays [4,12,13]. *H. pylori* colonizes the gastric tissues of more than half of the world’s population, causing chronic infections that can lead to gastritis, ulcers, and gastric cancers [14,15,16,17]. Current treatments for *H. pylori* infections have significantly decreased in efficacy globally [18], leading to the World Health Organization (W.H.O.) to categorize *H. pylori* as a high research priority due to alarming rates of antibiotic-resistant strains [19]. Community-based testing revealed that the frequency of infections that lead to gastric diseases and cancers are much higher in underserved communities in the US [20]. *H. pylori* relies on motility and chemotaxis for host colonization [12,21,22]. *H. pylori* has lophotrichous sheathed flagella that confer strong motility in viscous environments and a single chemotaxis system comprising four chemoreceptors [23,24]. It uses flagellar motility to migrate in media with soft agar concentrations below 0.6% [25].

At the start of this work, we noticed that *H. pylori* soft agar colonies form gaps when they encounter each other. Given the myriad reasons for gap formation, we sought to understand the basis in *H. pylori*. We examined various parameters that might contribute to gap formation including agar concentration, media composition, and bacterial genetic determinants. Our results support that agar and nutrient concentrations influence the rate of *H. pylori* colony migration, as observed previously [4,25], but these variables did not alter gap formation. By exploring quorum-sensing mutants and screening a transposon library, we found that gap formation is influenced by genetic factors. Our combined results suggest that the *H. pylori* soft agar gaps may be formed based on cell–cell recognition, a new idea that paves the way to understand this phenomenon.

## 2. Materials and Methods

### 2.1. Bacterial Strains and Growth Conditions

For this work, the following strains were used: *H. pylori* wild-type G27 [26], G27 pTM115 expressing GFP (KO446) [24], 26695 [27], PMSS1 [28], and SS1 [29]. The following mutants were used, all in the G27 or mG27 background: Δ*chePep* KO1332 [30], Δ*cheV2* KO1278 [31], Δ*cheV3* KO1279 [32], Δ*tlpA* KO740 [33], Δ*tlpB* KO583 [34], Δ*tlpC* KO568 [33]*, *Δ*tlpD* (KO#1006) [35], and Δ*luxS* (KO1791). For mutant isolation, a G27-based transposon library was used [36]. *H. pylori* strains were grown on solid media consisting of Columbia Horse Blood Agar (CHBA) containing 5% (*v*/*v*) defibrinated horse blood (Hemostat Labs, Dixon, CA, USA), 0.2% *w*/*v*-*β* cyclodextrin, 10 µg of vancomycin per mL, 5 µg of cefsulodin per mL, 2.5 U of polymyxin B per mL, 5 µg of trimethoprim per mL, and 8 µg of amphotericin B per mL (all chemicals from Thermo Fisher Waltham, MS, USA or Gold Biotech St. Louis, MO, USA). For growth in liquid media, Brucella broth (BB) (Thermo Fisher) plus 10% (*v*/*v*) heat-inactivated fetal bovine serum (HI-FBS) were used (BB10). For antibiotic resistance marker selection, bacterial media were supplemented with 25 µg of chloramphenicol (Cm) per mL or 15 µg of kanamycin (Km) per mL. For long-term storage, strains were frozen at −80 °C in brain heart infusion media supplemented with 10% HI-FBS, 1% (*w*/*v*) β-cyclodextrin, 25% glycerol, and 5% dimethyl sulfoxide. All *H. pylori* cultures were grown at 37 °C under microaerobic conditions of 5% O_2_, 10% CO_2_, and 85% N_2_.

### 2.2. Soft Agar Motility Assays

Soft agar motility assays were conducted with plates that were prepared using Brucella broth (BB) media with 2.5% HI-FBS and solidified with 0.35% (*w*/*v*) bacto-agar (Fisher Scientific) for standard soft agar plates, or different agar or BB amounts when examining these variables. Soft agar plates were inoculated by touching bacterial growth from CHBA plates using a sterile micropipette tip and inserting into the soft agar. Plates were incubated in microaerobic conditions at 37 °C for 6–10 days. Distances between the inoculation point and the colony edges were measured using a ruler. Soft agar plates were visualized using Microtek Scan Maker i900 (Cerritos, CA, USA) or Biorad Chemidoc MP Imaging System with White Epi Illumination setting on the UV Transilluminator (Hercules, CA, USA).

### 2.3. Isolation of Tn-7 Mutants Without Clear Gap Formation Phenotype

To isolate mutants that did not form soft agar gaps, a G27 transposon pool of mutants were cultured on CHBA for 3 days in microaerobic conditions. Soft agar plates were inoculated at two points 1 cm apart from each other. After incubating the plates for six days, the expanded colonies were partitioned to three zones: the non-competing front, the competing front, and the merge region, as shown in Figure 6A(i). Bacteria from each region were collected by poking sterile pipette tips into the agar at that region, serially diluted, and re-plated on Cm-supplemented CHBA plates to obtain single colonies. Twenty single colonies were collected from each of the three zones. These individual isolates were then plated again on standard soft agar to test behavior. Isolates that exhibited loss of gap formation were termed “merging isolates” and were stored at −80 °C for further analysis.

### 2.4. Identification of Transposon Insertion Sites

To identify the location of transposon insertion sites in the merging isolates, genomic DNA was extracted using the Wizard Genomic Prep Kit (Promega, Madison, WI, USA). Nested PCR was conducted on the extracted DNA. For the first round of PCR, amplification was carried out with a random primer with a constant tail region plus a transposon-specific primer, called CAT Tn7N and CAT Tn7S listed in Appendix A [36]. For the second round of PCR, amplification was carried out with a primer specific to the transposon plus a primer complementary to the constant tail region of the random primer, primers CAT Tn7 N2 and CAT Tn7 S2. Therefore, the final PCR products contain a portion of the transposon and the surrounding genomic information. These PCR products were then sent for DNA sequencing (Azenta, South Plainfield, NJ, USA) and analyzed by mapping to the G27 genome using NCBI Nucleotide Blast.

### 2.5. Genomic Transformations

To retransform the transposon mutants into a clean background, genomic DNA was isolated from the *alpA*::Tn-7 using the Wizard Genomic Prep Kit (Promega). *H. pylori* G27 WT was grown on CHBA for three days and then collected into 200 µL of BB10, such that the OD_600_ range was between 2 and 5, following the protocol in [37]. In total, 100 ng of genomic DNA was added to the bacterial suspension and incubated at room temperature for 5 min. The bacterial suspension was plated on CHBA for 24 h to allow recovery before being re-suspended in 300 µL of brucella and plated on CHBA plates supplemented with chloramphenicol. After 3–5 days of growth, single colonies were harvested, re-colony purified, and then either stored at −80 °C or used for DNA extraction.

### 2.6. Microscopy

To examine bacteria present in soft agar assays, samples were collected from the gaps by a pipet tip and visualized by phase-contrast microscopy at 200×, 400×, and 1000× magnification, using a Nikon ECLIPSE E600 microscope (Tokyo, Japan) with a Hamamatsu C7472-95 digital camera (Bridgewater, NJ, USA).

### 2.7. Growth Curves

*H. pylori* strains were cultured overnight (shaking) in BB10 for 12–14 h. Overnight cultures were back-diluted to OD_600_ of 0.1 using Brucella broth supplemented with 2.5% HI-FBS to match the components in the soft agar assays. Growth assays were completed with CLARIOstar*^Plus^* Microplate Reader (BMG LABTECH, San Francisco, CA, USA) under shaking, microaerobic conditions of 5% O_2_, 10% CO_2_, and 85% N_2_ at 37 °C. Four biological replicates per strain were conducted, and OD_600_ was measured hourly for 20–25 technical replicates for each time point over the course of 24 h.

### 2.8. Clear Gap Material Assessment

To assess whether the gaps had growth inhibitory compounds, gap material was extracted from between G27 WT colonies after 8 days of soft agar incubation. A P200 pipette man set at 100–200 µL was inserted about 3 mm into the agar, and material was collected by slow and consistent retraction to extract liquid. Typically, 8–10 plates were used to collect 1.5 mL. To this material, an equal volume of culture of the strain to be tested was added at a back-diluted OD_600_ of 0.1 in fresh BB10. This mixture was incubated in microaerophilic conditions with shaking for 8 h. Treated strains included G27 GFP+, which is kanamycin-resistant. Control cultures were treated with either 1.5 mL of material extracted from uninoculated soft agar or with BB10 only. Aliquots from treated cultures were serially diluted and plated on CHBA with either Kan or Cm; this step ensured that only that test strains would grow. CHBA plates were incubated for 3 days and counted for colony-forming units.

## 3. Results

### 3.1. H. pylori Ceases Soft Agar Migration and Forms a Gap When Two Colonies Meet

This project was initially started from the observation that *H. pylori* on long-incubated soft agar plates migrated from the inoculation points and saturated all the available media but produced a pattern of clear gaps in the soft agar that separated the colonies (Figure 1A). This result suggested that colonies from different inoculation points failed to intersect or merge. We next characterized the soft agar behavior in more detail using a wild-type *H. pylori* G27 strain that expresses GFP from the pTM115 plasmid [24]. *H. pylori* GFP+ was inoculated on a standard *H. pylori* soft agar plate at two points, one centimeter apart from each other. The plates were incubated and analyzed over the course of six days, measuring the distances migrated away from (non-competitive) or toward the other colony (competitive). The bacteria displayed little migration at days 1–2, as noted previously [25], but displayed visible outward migration from day 3 on (Figure 1B(i–iii)). During days 1–2, the distance migrated by the competitive and non-competitive front from the inoculation points were similar (Figure 1C). After 3 days, the two fronts started to show different behavior that was statistically significant, with the competitive bacterial front migrating slower than the non-competitive front (Figure 1C). Over days 3–6, the non-competitive bacterial fronts continued migrating, the colonies developed an asymmetric shape (Figure 1B), and the difference in distances traveled by both fronts became increasing significant by the final day of the assay. We confirmed this behavior in other *H. pylori* strains that were all isolated from varied human infections including 26695 [27], PMSS1 [28], and SS1 [29], observing that they too formed gaps. These results show that several *H. pylori* strains respond to each other even when at a distance and slow their migration as they approach.

### 3.2. Formation of Colony Gaps Is Chemotaxis-Independent

Soft agar migration depends on chemotaxis and motility. We therefore explored how bacterial chemotaxis affected gap formation. *H. pylori* G27 mutants with partial chemotaxis defects (Δ*chePep*, Δ*cheV2*, Δ*cheV3*) were used for this analysis because they maintain partial ability to migrate in soft agar [30,31,32]. These strains still formed gaps (Figure 2A), with migration on the non-competing front expanding normally as indicated on panel A1 of Figure 2. Similarly, mutants with single chemoreceptor mutations (Δ*tlpA*, Δ*tlpB*, Δ*tlpC*, Δ*tlpD*) retain soft agar migration [33,34,35] and still produced gaps (Figure 2B). These results suggest that chemotaxis does not affect gap formation between *H. pylori* soft agar colonies.

### 3.3. H. pylori Colony Gap Formation Is Independent of Nutrient Level and Agar Percentages

We next explored whether media parameters influenced gap formation. One previously hypothesized reason for gap formation is the nutrient-based explanation, which theorizes that gaps form as nutrients decrease because there are insufficient nutrients to support growth [5]. This theory predicts that lower nutrients would result in increased gap sizes. We therefore varied the concentration of Brucella broth in the media. Decreasing the Brucella broth below normal (0.5× or 0.75×) or above (1.5×) changed the migration rate, as reported before [4,38], but had no effect on gap formation (Figure 3A). These results suggest that nutrient availability does not affect *H. pylori* gap formation.

Other variables known to affect gap formation are physical properties, such as agar percentage [6]. We therefore modified the percent of agar in the soft agar plates. Gaps were readily apparent at the competitive bacterial fronts at agar concentrations from 0.3% to 0.45% (Figure 3B). At the lowest percent, 0.25%, there was only a minimal gap (Figure 3B). This result suggested that gaps require a minimal agar percentage for either formation or stability. At higher agar percent, the colonies had low migration but still retained gaps (Figure 3B). These results indicate that agar concentration may affect gap stability but does not clearly alter gap formation.

### 3.4. Soft Agar Colony Gaps Contain Low Densities of H. pylori Cells with Spiral Morphology

The gaps formed between colonies lack the normal opaqueness associated with bacterial growth, so we next examined whether these regions contained bacteria and whether these bacteria were alive. We first attempted to image the bacteria directly in the agar, by pouring thin, soft agar onto microscope slides with the Petri dish as conducted previously [25], but these agar slides were too optically dense to see bacteria. So, samples were instead collected directly from the soft agar in either the gap or colony and imaged using phase-contrast microscopy (Figure 4). Low-magnification images showed that bacteria were present in the gaps at a low density, but the cells were too small to gather any morphological features (Figure 4A,B). At 1000×, cells in the gap had a spiral morphology (Figure 4A(iii)), while those within the middle of the colony showed non-spiral-shaped cells in aggregates (Figure 4B(iii)). The direct plating of gap material showed *H. pylori* growth (Figure 4C). These results suggest that the gaps contain a low density of *H. pylori* cells that are spiral in morphology, suggesting that they are potentially viable.

### 3.5. Assessing Inhibitory Compounds in Gap Regions

One possibility is that the gap region contains compounds that inhibit the accumulation of bacteria. This possibility was tested by exposing an experimental strain to material collected from gaps. Specifically, gaps were formed by wild-type *H. pylori* strain G27 (kanamycin-sensitive) for 8 days, and, then, material was collected from the gaps by pipetting. This material was then used to treat an *H. pylori* G27 strain that was kanamycin-resistant (*H. pylori* G27 pTM115). We then plated the treated samples on kanamycin CHBA, so that only the experimental strain would grow, comparing treated samples to those that were either left untreated or treated with uninoculated agar samples (experimental design shown in Appendix A). After treatment, the samples were serially diluted and plated (Figure 5A–C). Treatment with gap material resulted in an ~10-fold decrease in CFU counts compared to no treatment (Figure 5D), but control experiments showed that agar alone was slightly inhibitory. When comparing the two agar treatments to each other (±*H. pylori*), there was lower growth, but this decrease did not achieve statistical significance. These results suggest that there may be some inhibitory activity in the gap region, but it is not substantial or toxic.

### 3.6. H. pylori Gap Formation Has a Genetic Basis

We next analyzed whether there might be genetic determinants that affected gap formation. One *H. pylori* property that influences group behavior is quorum sensing via the auto-inducer 2 (AI-2) quorum-sensing system [39]. To test whether quorum sensing was involved in gap formation, we analyzed the behavior of *H. pylori* lacking the AI-2 producing enzyme, *luxS*. *H. pylori* Δ*luxS* mutants did not form gaps (Figure 6B), suggesting that gap formation depended on bacterial properties including quorum sensing and was not solely eliminated in strains with transposons. To identify other genetic loci that controlled gap formation, we screened an *H. pylori* G27 mini-Tn7 transposon mutant pool [36]. As performed above, we incubated two inoculums of this pool one centimeter apart for six days, and samples were taken from the merging corners of the gap region to enrich for mutants that had lost the ability to form gaps. As controls, isolates were also collected from within the colony, at both the non-competing and competing fronts as defined in Figure 1A. These samples were single-colony purified and retested for gap formation, resulting in a collection of 20 merging mutants. Isolates from within the colony, from either the non-competing or competing fronts, retained the ability to form gaps (Figure 6A), but isolates from the gap region lost this ability, creating colonies that merged (Figure 6A). To characterize whether the “merge” (loss-of-gap) phenotype was associated with the Tn7-targeted locus, we isolated genomic DNA from these merge isolates and used this DNA to transform a fresh G27 WT strain to chloramphenicol resistance. These recreated strains also lost gap formation, suggesting that the Tn7-mutated locus conferred the merge phenotype (Figure 6B). These results suggested that the transposons had interrupted genes that were required for gap formation.

To identify the mutant loci, nested PCR was used to identify the disrupted genes in 14/20 single-colony Tn7 mutants with the merge phenotype. The disrupted genes included those coding for the AlpA and AlpB outer membrane proteins, two cysteine-rich proteins, the HpyAIV type II restriction enzyme, phosphatidyl glycerophosphate synthase, and four hypothetical proteins (Table 1). These results suggest that *H. pylori* soft agar colony gap formation has a genetic basis, but it appears that the loss of any of several loci can confer a merging phenotype.

To gain insight into the underlying reason that the Tn mutants merged, we analyzed various aspects of their behavior. These mutants retained merging behavior under varying nutrient and agar concentrations (Appendix A). We observed that several of these mutants had elevated migration rates (Appendix A). We evaluated their growth rates but found that this was not altered (Appendix A). To explore whether a fast migration rate could have potentially contributed to the loss of the gaps, e.g., perhaps by allowing the mutant strains to overcome physical resistance of the soft agar, we examined a known mutant with a documented elevated migration rate, *H. pylori* G27 Δ*pilO* [13]. This mutant, however, retained a normal ability to produce gaps between colonies, suggesting that fast migration may facilitate but is not sufficient to cause merging (Appendix A).

## 4. Discussion

In this work, we present findings that *H. pylori* migrating colonies do not merge and instead form low-density bacterial regions that appear as gaps, a phenomenon observed in other microbes. The *H. pylori* gaps are independent of chemotaxis, nutrient levels, or agar concentrations. They also do not have detectable soluble anti-microbial properties. Instead, our data suggest that these gaps form as a consequence of *H. pylori* physiological properties, as evidenced by the finding of *H. pylori* mutants that lose the ability to form gaps. This work may have implications for the observed behavior of *H. pylori* in vivo, in which one infection blocks colonization by a second *H. pylori* strain [11,22,24].

Previous work has shown that soft agar colonies of some bacterial species merge, while others form gaps and avoid direct colony-to-colony contact [5]. In some species, the phenomenon of merging or gap formation between competing bacterial colonies has been correlated with nutrient availability [5,7] or agar percentage [5,6]. Studies with *Salmonella* Typhimurium linked the “slow down” of migration speed to excess nutrients, which caused a decrease in bacterial front velocity and the creation of gaps, with cells documented to switch to a less motile state [5]. *B. subtilis* colonies, in comparison, merge under low-nutrient conditions, where migration speeds were faster than under nutrient-depleted conditions [7]. Gap formation in *H. pylori*, however, is not affected by nutrient levels. Specifically, elevating or lowering Brucella broth concentrations (Figure 3). In line with this finding, partial chemotaxis mutants still formed gaps in soft agar (Figure 2). Altogether, these results suggest that *H. pylori* gap formation is largely independent of media composition and chemotactic response.

We also explored whether the gaps between colonies might contain a potential secretion that decreases bacterial density. We collected gap material and treated fresh *H. pylori*, but we found that the gaps slightly decreased bacterial numbers; however, this decrease was not statistically significant (Figure 4). This finding suggested that gap formation is not due to cell growth inhibition via the secretion of toxic compounds.

To gain ideas for how gap formation occurs in *H. pylori*, we examined quorum-sensing mutants and isolated transposon mutants that do not produce gaps. Quorum-sensing defective *luxS* mutants were the first that we observed to lose gap formation (Figure 6). This finding strongly supported the idea that gap formation is influenced by bacterial properties, in this case, the production of AI-2. Consistent with this idea, our transposon mutant screen identified 10 transposon insertions that lost gap formation. These disrupted loci mapped to a variety of cellular processes, including outer membrane proteins, cysteine-rich proteins, metabolic proteins, restriction enzymes, and hypothetical proteins. All these proteins play important roles in *H. pylori* biology.

AlpAB are two outer membrane proteins characterized as adhesins that facilitate adherence between *H. pylori* and host–cell laminin, and they also contribute to outer membrane vesicles in *H. pylori* biofilms [40,41,42,43,44]. The *alpA-alpB* operon was previously found to play a role in colonization in gerbils [42], mice [40], guinea pigs [45], and human epithelial cells [44,46], and it triggers a host immune response [40,47]. AlpA and AlpB contain 518 amino acids and are highly similar [40], functionally dependent on each other [40] and expressed in similar amounts in 200 clinical *H. pylori* isolates from human gastric biopsies [41]. Based on the locations of the transposon insertions, the region upstream the *alpAB* operon and *alpA* was disrupted, indicating that the merge phenotypes we have observed may be due to the loss of at least *alpA* and maybe *alpB*.

Another transposon disrupted the gene for the type II restriction endonuclease HpyAIV. In the *H. pylori* strain G27, HpyAIV was recently identified as a gene that is upregulated transcriptionally by 2.59-fold in biofilm cells relative to planktonic cells [48]. Another disrupted gene, *HPG27_1061*, has not been characterized in G27, but its homolog was found to be a secreted protein dependent on environmental salt concentrations [49,50]. There are limited studies on the hypothetical proteins, cysteine-rich proteins, and the phosphatidyl glycerophosphate synthase. We do not yet know the functions of these genes in gap formation, suggesting we have much to learn about the roles of these proteins in general.

An interesting gene that was required for gap formation was *luxS*, a gene that facilitates a cell density-based signaling mechanism via the production of Auto Inducer 2 (AI-2), a molecule used to regulate population growth amongst many bacterial species [51]. In *H. pylori*, AI-2 quorum sensing has been shown to affect the expression of many genes, some involved in motility [52,53], hypothetical proteins, endorestriction nucleases, and outer membrane proteins [54]. Thus, the role of *luxS* in gap formation could be due to any of several processes. Orthologs of some of the other identified genes, *HPG27_147* and *HPG27_412*, have been shown to be impacted by LuxS in other *H. pylori* strains [54]. These genes encode a possible secreted beta-lactamase and a phosphatidyl glycerophosphate synthase. Unlike many bacteria that use AI-2 as an attractant, *H. pylori* perceive this molecule as a chemorepellent, moving away from its source using the TlpB chemoreceptors [53,55]. Although *H. pylori* possesses a negative chemotactic response to AI-2, gap formation still occurred in *tlpB* mutants, suggesting that chemotactic responses are not a causative factor for gaps. AI-2 was recently proposed to reduce *H. pylori* adhesion to AGS cells by suppressing the expression of outer membrane proteins including AlpA and AlpB [56], and the suppression of adhesion in *H. pylori* biofilms leads to the direct inhibition of growth [55]. One possibility is that, under high cell density, these adhesins that are regulated by quorum sensing may facilitate binding to specific substrates or structures within the soft agar, indicating that spatial segregation may be maintained through specific molecular mechanisms. Though virtually all *H. pylori* strains contain genes encoding LuxS, AlpA, and AlpB, their contribution to spatial organization between bacterial populations and potential binding to specific substrates or structures has yet to be fully elucidated.

Altogether, our results suggest that *H. pylori* forms gaps on soft agar due to bacterial-intrinsic properties. We favor a model in which quorum sensing and other *H. pylori* genes, e.g., *alpAB*, create a physiological state that leads to gaps, which may involve a combination of changes to adhesive properties, the lowering of growth, or possibly even inhibiting growth of the encroaching colony. One possibility is that *H. pylori* detects the encroaching colony as “foreign” and has the ability to inhibit these cells. Although clonal populations are genetically identical, recent research in other pathogenic bacteria with isogenic strains has highlighted the role of sociomicrobiology in biofilms [10] and in vivo [39,57]. For example, isogenic *P. aeruginosa* strains can outcompete each other in murine lungs [57], and *S. pneumoniae* engages in fratricide to outcompete isogenic strains in murine nasal cavities [39]. Multiple studies have documented competition dynamics between *H. pylori* inoculums that affect gland occupation in vivo [11,24]; however, kin selection and competition mechanisms have not been well characterized in *H. pylori.* Our experiments allude to a possible non-secreted spatial regulation of competing *H. pylori* populations, although our data cannot rule out the possibility that these interactions confer benefits, e.g., enhanced antibiotic heteroresistance between resistance and sensitive populations [57]. Further understanding *H. pylori* social distancing mechanisms could reveal mechanisms that regulate *H. pylori* populations in the host and help us better understand previously documented gland occupation dynamics in vivo.

## Figures and Tables

**Figure 1 microorganisms-13-01087-f001:**
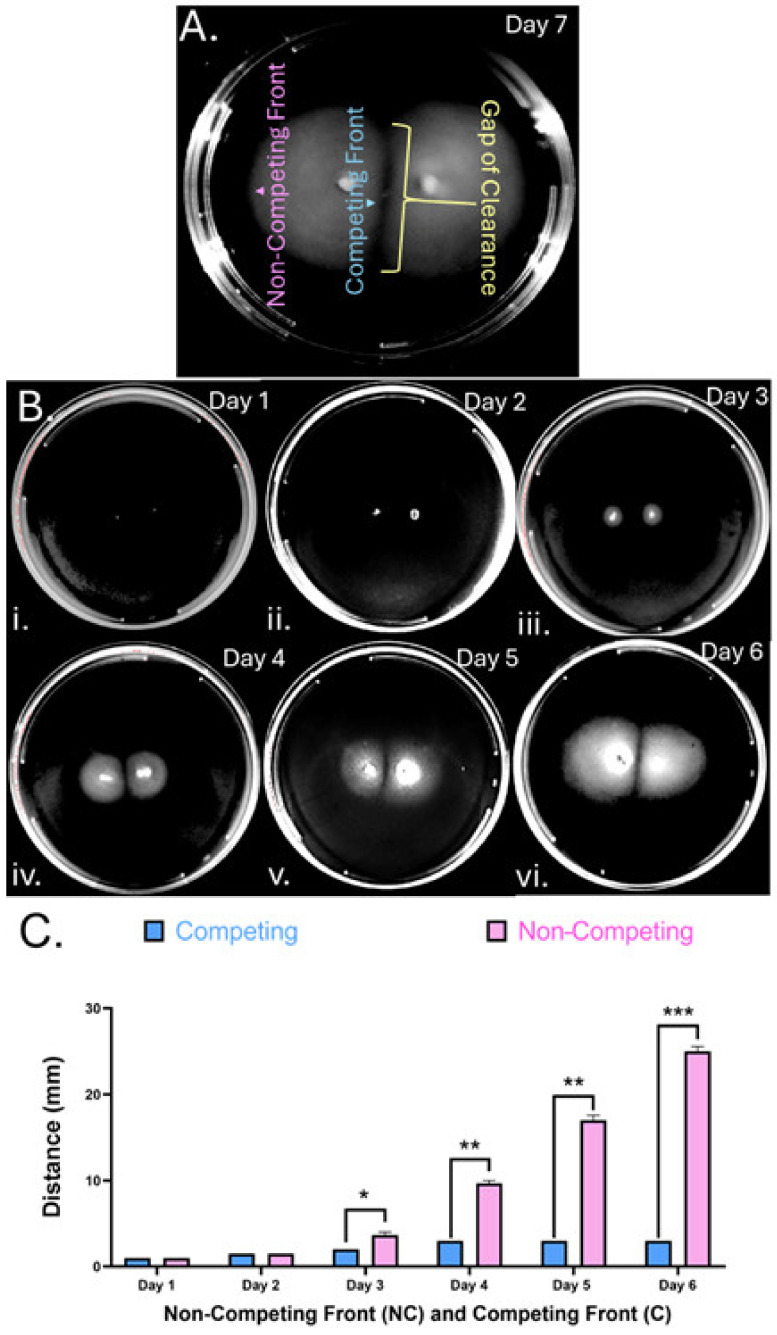
***H. pylori* soft agar inoculations produce gaps between adjacent colonies:** *H. pylori* strain G27 pTM115 (GFP+) inoculated in standard *H. pylori* soft agar media, incubated under microaerobic conditions, and imaged at the indicated times. (**A**) At 7 days of incubation, soft agar inoculations have three zones: non-competing front, competing front, and gap. (**B**) The plate shown in panel A is imaged days one–six. (**C**) The distances migrated by the competing (**C**) and non-competing (NC) fronts from the inoculation point are measured daily over six days for 4 biological replicates. Shown are averages with error bars representing SEM. Biological replicates are compared statistically using paired T-test analysis with significance indicated by * (*p*-value < 0.05), ** (*p*-value < 0.01), and *** (*p*-value < 0.001).

**Figure 2 microorganisms-13-01087-f002:**
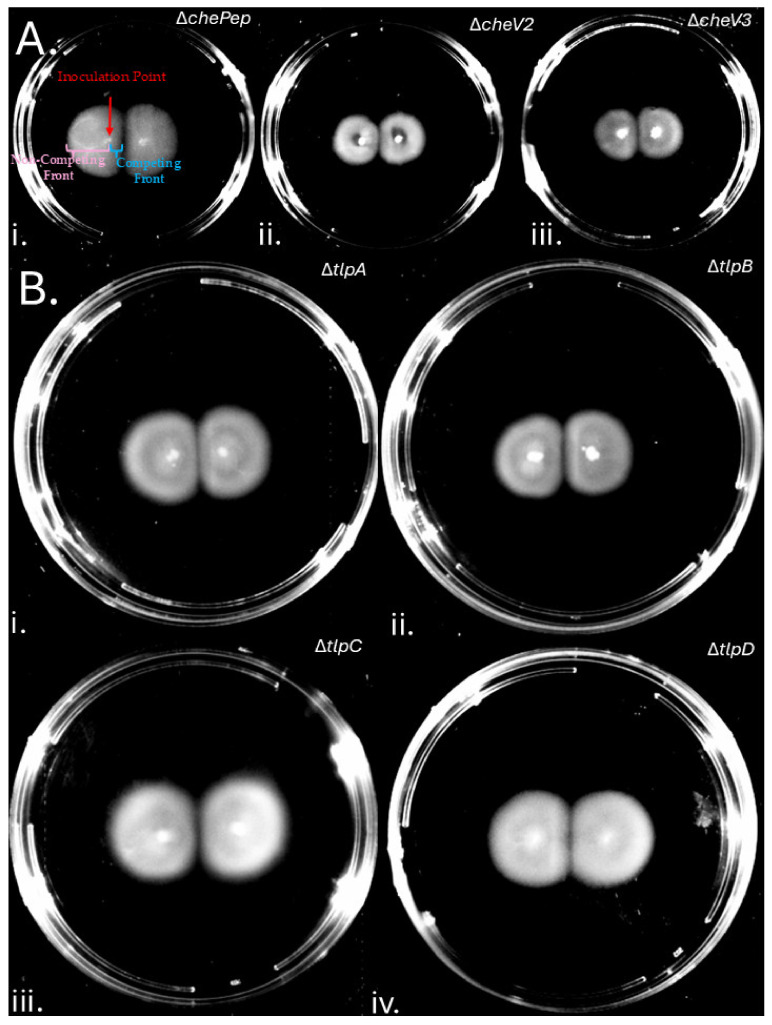
***H. pylori* soft agar colony gaps are independent of chemotaxis:** Partial chemotaxis and chemoreceptor mutants in strain G27 inoculated in soft agar and imaged after 9 days (panel **A**) or 6 days (panel **B**). (**A**) Chemotaxis signaling mutants lacking (i) Δ*chePep*, (ii) Δ*cheV2*, (iii) Δ*cheV3*. (**B**) Chemotaxis receptor mutants (i) Δ*tlpA*, (ii) Δ*tlpB*, (iii) Δ*tlpC*, (iv) Δ*tlpD*. Images are representative of four biological replicates per mutant.

**Figure 3 microorganisms-13-01087-f003:**
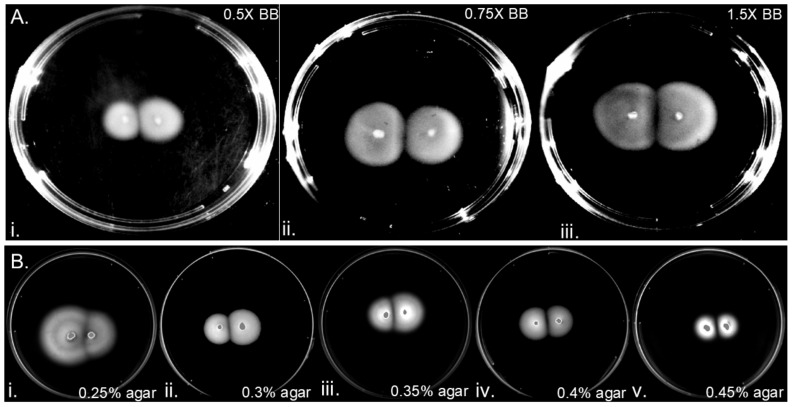
**Gap formation is independent of soft agar nutrient and agar components:** Soft agar plates inoculated in two places 1 cm apart with H. pylori G27 GFP+ on plates with varying Brucella broth (BB) (**A**) or agar levels (**B**); (**A**) (i) 0.5XBB for 7 days, (ii) 0.75XBB for 7 days, and (iii) 1.5XBB for 9 days; (**B**) (i) 0.25%, (ii) 0.3%, (iii) 0.35%, (iv) 0.4%, and (v) 0.45% agar.

**Figure 4 microorganisms-13-01087-f004:**
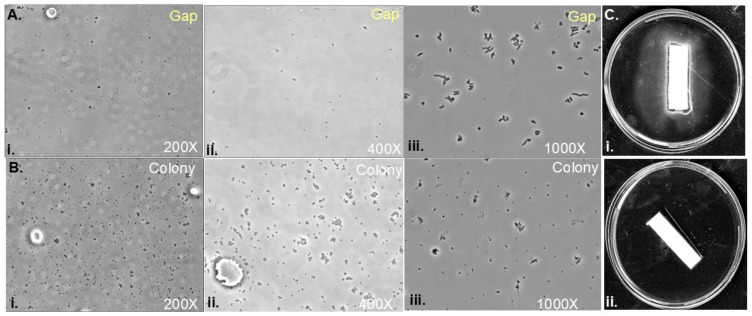
**Gaps of clearance contain *H. pylori* with spiral morphology at a low but non-zero density.** *H. pylori* G27 soft agar plates inoculated in two spots 1 cm apart and samples from the resulting colonies imaged after 6 days using microscopy. A 10 µL aliquot of soft agar taken from either (**A**) from within the gap region (top panels) or (**B**) from within the colony (bottom panels) and imaged with phase-contrast microscopy, at (i) 200× magnification, (ii) 400× magnification, or (iii) 1000× magnification. Images are indicative of three biological replicates. (**C**) Whatman paper is inserted into the clear gap of inoculated plates (i) and then plated to soft agar for 6 days, showing bacterial growth in comparison to a negative control, a Whatman paper that is inserted in uninoculated agar (ii).

**Figure 5 microorganisms-13-01087-f005:**
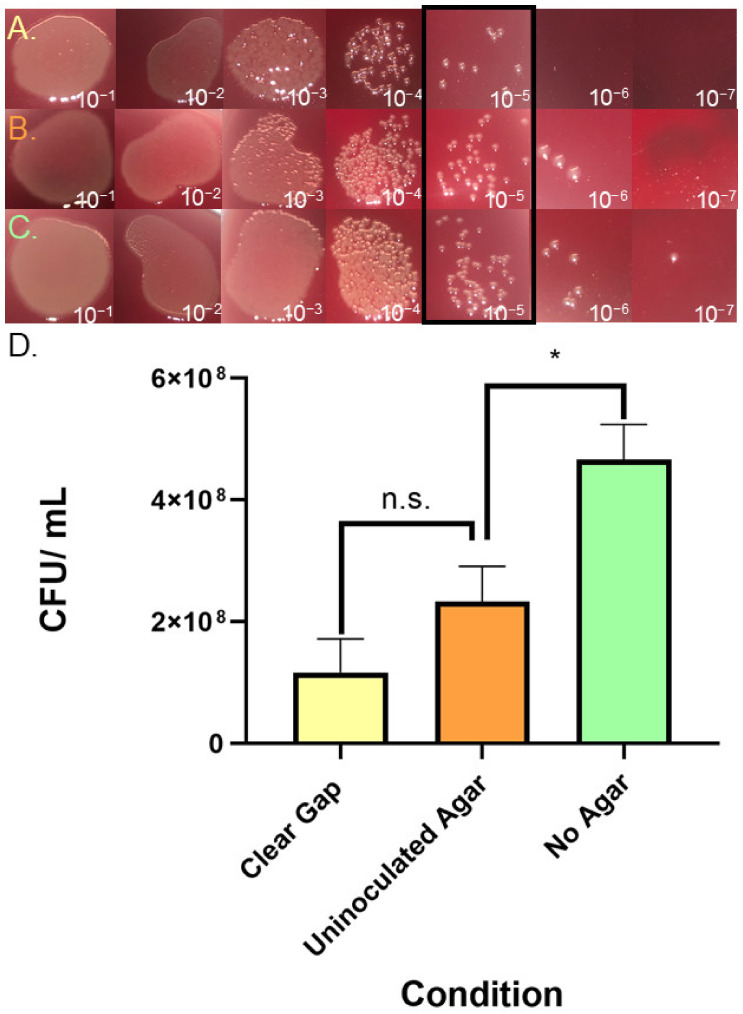
**Analysis of clear gap effect on growth:** *H. pylori* G27 pTM115 (Kan^R^) liquid cultures are incubated for seven hours in microaerobic conditions with material collected from (**A**) clear gap extracted from between WT *H. pylori* G27 colonies (**B**), uninoculated agar, or (**C**) no treatment control. After treatment, samples are serially diluted and spot-plated. (**D**) CFU data calculated from 3 biological replicates; shown are averages with error bars representing SEM of samples treated as in (**A**–**C**). (n.s., *p*-value = 0.0646; *, *p*-value < 0.05) using Student’s *t*-test analysis.

**Figure 6 microorganisms-13-01087-f006:**
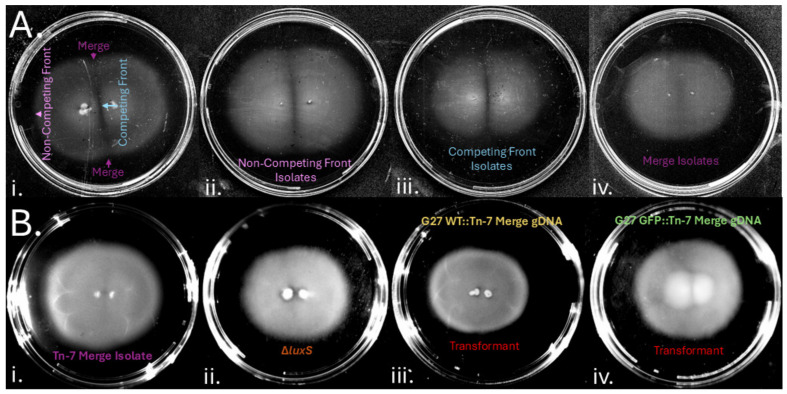
**Isolation of genetic mutations that cause *H. pylori* to lose gap formation:** (**A**) A *H. pylori* G27 Tn7 pool [36] is plated at positions 1 cm apart for 6 days on standard soft agar (i). Isolates are taken from the non-competitive front, the competitive front, and the gap regions. Example phenotypes observed in Tn7 mutants isolated from within the colony at the non-competitive front (ii), the competitive front (iii), or from the gap region (iv). (**B**) Merging phenotypes observed in (i) Tn-7 mutants (ii) Δ*luxS* and the merging phenotypes can be recapitulated when transforming Tn-7 genomic DNA into (iii) G27 WT or (iv) G27 GFP+.

**Table 1 microorganisms-13-01087-t001:** Genes with Tn-7 insertion in merging mutants.

Gene Disrupted by Tn-7	Gene Function	Number of Isolates
*HPG27_843*	Outer membrane protein HopC/AlpA	3
Intergenic *HPG27_843&844*	Outer membrane protein HopCB/AlpAB locus	2
*HPG27_147*	Cysteine-rich protein D	1
*HPG27_412*	Phosphatidyl glycerophosphate synthase	1
*HPG27_413*	Hypothetical protein	1
*HPG27_462*	Hypothetical protein	1
*HPG27_1061*	Cysteine-rich protein X	1
*HPG27_1062*	Hypothetical protein	1
*HPG27_1299*	Type II endorestriction nuclease HpyAIV	2

## Data Availability

The original contributions presented in this study are included in the article/Appendix A. Further inquiries can be directed to the corresponding authors.

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
