# Peer review of "Genetic Basis of Gap Formation Between Migrating *Helicobacter pylori* Colonies in Soft Agar Assays"

_microorganisms, 2025, doi:10.3390/microorganisms13051087_

Round 1

Reviewer 1 Report

Comments and Suggestions for Authors

Elshenawi et al show that Helicobacter pylori colonies growing on soft agar will not merge with each other, but stop to extend at the edge of contact, while non-contact edges continue to extend. Gap formation is associated with few cells being present within the gap, and is independent of motility or medium composition. Using a transposon screen, the authors find several genes whose absence leads to loss of gap formation.

This is a very interesting study and nice to read manuscript. I have only a few comments.

Fig. 2 looks as though the non-competing edge of several chemotaxis mutants also stops extending as colonies meet. Can the authors rule this out by performing distance measurements as in Fig. 1C? Or maybe the shown colonies are exceptions, then replace them with the more representative examples.

Line 262 “by pipetting” can this procedure be explained a bit more in detail, I have a hard time picturing how any liquid could be pipetted off a soft agar plate.

Section 3.6: I am not clear if mutants can only be found by isolating cells from gaps. A) does this procedure in wild type cells generate gap-less variants? B) Was it impossible to obtain gap-less colonies directly from the Tn collection?

And how many Tn mutant strains were tested?

Line 316: do the authors mean to say that it takes several mutations accumulating in cells to get a gap-less strain?

Line 334 – better “detectable” than “substantial”

Line 345 – “in contrast” – I do not see any contrast between the sentences

Line 347 – “amounts” is not really fitting

364 – outer membrane

372 - dependent on each other functionally  “functionally dependent on each other”

Author Response

Dear Editor,

Thanks for obtaining the clear and insightful reviews, we have addressed all the

comments as you can see below.

Thank you, Karen Ottemann and Skander Hathroubi

Reviewer 1

Fig. 2 looks as though the non-competing edge of several chemotaxis mutants also stops

extending as colonies meet. Can the authors rule this out by performing distance

measurements as in Fig. 1C? Or maybe the shown colonies are exceptions, then replace them

with the more representative examples.

Response: Thank you for this comment. To clarify, the non-competing fronts do not stop

extending. To make this result more clear, we have modified the figure to include measurements

on panel A (panel shown below) and also made this clear in the text at line 212 by adding the

phrase “..with migration on the non-competing front expanding normally as indicated on panel A1

of Fig. 2”.

Line 262 “by pipetting” can this procedure be explained a bit more in detail, I have a hard

time picturing how any liquid could be pipetted off a soft agar plate.

Response: Thank you for this question. We added more information to the methods at line 161:

“A P200 pipetteman set at 100-200uL was inserted about 3 mm into the agar, and material

collected by slow and consistent retraction to extract liquid. Typically, 8-10 plates were used to

collect 1.5 ml.”

Section 3.6: I am not clear if mutants can only be found by isolating cells from gaps. A) does

this procedure in wild type cells generate gap-less variants? B) Was it impossible to obtain

gap-less colonies directly from the Tn collection?

Response: Interesting questions. We did not try to isolate mutants from non-transposon/WT

libraries, but we imagine they could be. We know that mutants do not have to have the transposon,

as the luxS mutant pictured in Figure 6BII was not isolated from the Tn7 pool, a point we havehighlighted by adding to the text at line 300 “suggesting gap formation depended on bacterial

properties including quorum sensing and was not solely eliminated in strains with transposons.”

(B) As to obtaining mutants directly from the Tn collection, this seems challenging as they are

likely at a low abundance, so we did not attempt this.

And how many Tn mutant strains were tested?

Response: Thanks for pointing out this omission. We collected a total of 20 isolates from the

10,000 Tn 7 Mutant pool that could merge and had a loss of the gap, but with the nested PCR we

were only able to identify the transposon insertions in 14 of those mutants. This information has

been clarified at line 307 by adding “resulting in collection of 20 merging mutants” and at line 329

“To identify the mutant loci, nested PCR was used to identify the disrupted genes in 14/20 single

colony Tn7 mutants with the merge phenotype”

.

Line 316: do the authors mean to say that it takes several mutations accumulating in cells to

get a gap-less strain?

Response: Sorry for any confusion, we realized the original statement was ambiguous. The

original statement was “These results suggest that H. pylori soft agar colony gap formation has a

genetic basis, with multiple genes implicated” so we’ve now modified this at current line 332 to

“These results suggest that H. pylori soft agar colony gap formation has a genetic basis, but it

appears that loss of any of several loci can result in a merging phenotype”

.

Line 334 – better “detectable” than “substantial”

Response: Changed the wording as suggested.

Line 345 – “in contrast” – I do not see any contrast between the sentences

Response: Changed “in contrast” to “in comparison”.

Line 347 – “amounts” is not really fitting

Response: Changed amounts to levels

364 – outer membrane

Response: Added a space between outer and membrane

372 - dependent on each other functionally “functionally dependent on each other”

Response: Changed it as suggested

Reviewer 2 Report

Comments and Suggestions for Authors

I believe that the original article entitled "Genetic Basis of Gap Formation Between Migrating Helicobacter pylori Colonies in Soft Agar Assays” is interesting and sheds new light on another biological phenomenon related to this bacterium. The inclusion of genetic studies, which allow us to delve deeper into this area, certainly deserves recognition in this context. To improve the quality of the manuscript, I would like to suggest the following amendments:

Line 41: "the expanding colonies can potentially run into each other" -> it is clearly not very scientific, please change it

Line 57 and 342: " Salmonella typhimurium" -> should be written as Salmonella Typhimurium (the full name stands for Salmonella enterica serovar Typhimurium and because of this when using the shorter version “Salmonella Typhimurium” is required)

Line 44 and 60: in line 44 Authors should write the full name of the bacterium (Helicobacter pylori) and in line 60 a shorter version (H. pylori) - as the full name was used previously

Lines 60-66: please add several more sentences about the pathogenicity or diseases produced by this bacterium; why it is so important to study this microbe?

Section 3.4: I believe that it is worth observing the morphology of bacteria from different sections of the colony using SEM in the future - then it will be possible to obtain precise information not only on the shape of cells but also their spatial organization. The cut fragment of agar with a fragment of the colony on its surface (or the section from the gap) should be thrown directly into an Eppendorf tube with an glutaraldehyde solution, then the sample should be passed through the alcohol series and sprayed with gold. After this the obtained biological sample can be viewed in SEM without disturbing the delicate structure of this sample.

Line 405: "AlpBtheir" -> a gap between the words is missing

Line 417-427: Of course, while there are studies showing competition between H. pylori strains, there are many that indicate numerous benefits resulting from the co-occurrence of H. pylori strains in one niche, e.g. the phenomenon of heteroresistance to antibiotics. I believe that it is worth expanding the discussion to include such considerations, especially since a clear answer regarding the function of creating gaps between colonies has not been established in the current article.

Author Response

Dear Editor,

Thanks for obtaining the clear and insightful reviews, we have addressed all the

comments as you can see below.

Thank you, Karen Ottemann and Skander Hathroubi

Reviewer 2

Line 41: "the expanding colonies can potentially run into each other" -> it is clearly not very

scientific, please change it

Response: Changed it to “colonies can be in close proximity to one another” at current line 41.

Line 57 and 342: " Salmonella typhimurium" -> should be written

as Salmonella Typhimurium (the full name stands for Salmonella enterica serovarTyphimurium and because of this when using the shorter version “Salmonella

Typhimurium” is required)

Response: Thank you for this correction, it has been changed as suggested

Line 44 and 60: in line 44 Authors should write the full name of the bacterium (Helicobacter

pylori) and in line 60 a shorter version (H. pylori) - as the full name was used previously

Response: Thank you for this correction, much appreciated, and we made the suggested change.

Lines 60-66: please add several more sentences about the pathogenicity or diseases produced

by this bacterium; why it is so important to study this microbe?

Response: Thank you for this comment, we added a couple of sentences highlighting the

significance of studying H. pylori and its impact/burden on public health at current line 66,

specifically “Current treatments for H. pylori infections have significantly decreased in efficacy

globally [18], leading to the World Health Organization (WHO) to categorize H. pylori has a high

research priority due to alarming rates of antibiotic resistant strains [19]. Community-based testing

revealed that the frequency of infections that lead to gastric diseases and cancers are much higher

in underserved communities in the US [20].

Section 3.4: I believe that it is worth observing the morphology of bacteria from different

sections of the colony using SEM in the future - then it will be possible to obtain precise

information not only on the shape of cells but also their spatial organization. The cut

fragment of agar with a fragment of the colony on its surface (or the section from the gap)

should be thrown directly into an Eppendorf tube with an glutaraldehyde solution, then the

sample should be passed through the alcohol series and sprayed with gold. After this the

obtained biological sample can be viewed in SEM without disturbing the delicate structure

of this sample.

Response: We thank the reviewer for this interesting idea for future experiments, we will give that

a try and agree that this question is for the future but not this work.

Line 405: "AlpBtheir" -> a gap between the words is missing

Response: Thank you for catching that, we have corrected it.

Line 417-427: Of course, while there are studies showing competition between H. pylori

strains, there are many that indicate numerous benefits resulting from the co-occurrence

of H. pylori strains in one niche, e.g. the phenomenon of heteroresistance to antibiotics. I

believe that it is worth expanding the discussion to include such considerations, especially

since a clear answer regarding the function of creating gaps between colonies has not been

established in the current article.

Response: We appreciate that the reviewer highlighted this point, we totally agree. We added a

line to the discussion to represent this idea at line 433, as “…although our data cannot rule out the

possibility that these interactions confer benefits, e.g. enhanced antibiotic heteroresistance between

resistance and sensitive populations.”

Reviewer 3 Report

Comments and Suggestions for Authors

This is an interesting manuscript looking into gap formation by H. pylori when cultured in soft agar. The authors did an extensive set of experiments to determine the nature of these avoidance features. They have identified a set of proteins likely to play a role in avoidance with other H. pylori. Their results suggest that in nature H. pylori strains do not mix due to neighboring recognition.

Minor comments

  • Figure 4 is difficult to interpret in terms of the described phenotypes. Nevertheless, phenotypic differences occur between gaps and individuals from the center colonies.
  • Minor grammatical errors

The authors described that their findings may have implications for the observed behavior of H. pylori, where one infection blocks colonization by a second H. pylori strain. While this statement is appropriate based on their findings, there are a couple of questions:

  • Have the authors expanded these findings to naturally isolated H. pylori?
  • How can we explain the high genetic exchange known to happen in vivo?

Author Response

Dear Editor,

Thanks for obtaining the clear and insightful reviews, we have addressed all the

comments as you can see below.

Thank you, Karen Ottemann and Skander Hathroub

Figure 4 is difficult to interpret in terms of the described phenotypes. Nevertheless,

phenotypic differences occur between gaps and individuals from the center colonies.

Response: Thank you to the reviewer for pointing this out, we modified the text to clarify that we

did not expect people do see the cell shapes at low mag, and only at 1000X by adding at line 246

“Low magnification images showed that bacteria were present in the gaps at a low density, but the

cells were too small to gather any morphological features (Fig. 4A, B). At 1000X, cells in the gap

had a spiral morphology (Fig. 4Aiii) while those within the middle of the colony showed non-

spiral shaped cells in aggregates (Fig. 4Biii).”

Minor grammatical errors

Response: Thanks for pointing this out, we read the entire manuscript over and made corrections

as needed.

The authors described that their findings may have implications for the observed behavior

of H. pylori, where one infection blocks colonization by a second H. pylori strain. While this

statement is appropriate based on their findings, there are a couple of questions (1) Have the

authors expanded these findings to naturally isolated H. pylori? (2) How can we explain the

high genetic exchange known to happen in vivo?

Response: We thank the reviewer for pointing these issues out. First, we analyzed several H. pylori

strains as indicated in the text at line 188 (26695, PMSS1, and SS1), which have all been isolated

from human infections; we mentioned the isolation information by updating the sentence at line

190 with “We confirmed this behavior in other H. pylori strains that were all isolated from varied

human infections including 26695 [27], PMSS1 [28], and SS1 [29], observing that they too formed

gaps “
